# Hammerstein–Wiener Motion Artifact Correction for Functional Near-Infrared Spectroscopy: A Novel Inertial Measurement Unit-Based Technique

**DOI:** 10.3390/s24103173

**Published:** 2024-05-16

**Authors:** Hayder R. Al-Omairi, Arkan AL-Zubaidi, Sebastian Fudickar, Andreas Hein, Jochem W. Rieger

**Affiliations:** 1Applied Neurocognitive Psychology Lab, Carl von Ossietzky Universität Oldenburg, 26129 Oldenburg, Germany; hayder.r.alomairi@uotechnology.edu.iq (H.R.A.-O.); arkan.al-zubaidi@uni-oldenburg.de (A.A.-Z.); 2Department of Biomedical Engineering, University of Technology—Iraq, Baghdad 10066, Iraq; 3Cluster of Excellence Hearing4all, Carl von Ossietzky Universität Oldenburg, 26129 Oldenburg, Germany; 4Assistance Systems and Medical Device Technology, Carl von Ossietzky University Oldenburg, 26129 Oldenburg, Germany; sebastian.fudickar@uni-oldenburg.de (S.F.); andreas.hein@uni-oldenburg.de (A.H.); 5Institute for Medical Informatics, University of Lübeck, 23562 Lübeck, Germany

**Keywords:** functional near-infrared spectroscopy fNIRS, motion artifact, multi-channel IMU, accelerometer, gyroscope, motion correction, NIRS signal improvement

## Abstract

Participant movement is a major source of artifacts in functional near-infrared spectroscopy (fNIRS) experiments. Mitigating the impact of motion artifacts (MAs) is crucial to estimate brain activity robustly. Here, we suggest and evaluate a novel application of the nonlinear Hammerstein–Wiener model to estimate and mitigate MAs in fNIRS signals from direct-movement recordings through IMU sensors mounted on the participant’s head (head-IMU) and the fNIRS probe (probe-IMU). To this end, we analyzed the hemodynamic responses of single-channel oxyhemoglobin (HbO) and deoxyhemoglobin (HbR) signals from 17 participants who performed a hand tapping task with different levels of concurrent head movement. Additionally, the tapping task was performed without head movements to estimate the ground-truth brain activation. We compared the performance of our novel approach with the probe-IMU and head-IMU to eight established methods (PCA, tPCA, spline, spline Savitzky–Golay, wavelet, CBSI, RLOESS, and WCBSI) on four quality metrics: SNR, △AUC, RMSE, and R. Our proposed nonlinear Hammerstein–Wiener method achieved the best SNR increase (*p* < 0.001) among all methods. Visual inspection revealed that our approach mitigated MA contaminations that other techniques could not remove effectively. MA correction quality was comparable with head- and probe-IMUs.

## 1. Introduction

Functional near-infrared spectroscopy (fNIRS) noninvasively measures changes in oxyhemoglobin (HbO) and deoxyhemoglobin (HbR) levels in the cortex with near-infrared light (e.g., 760 and 850 nm) [1]. To this end, light sources and detectors are attached to the head. The intensity of light arriving at a detector situated several centimeters away from the source is measured and then converted to concentration changes of HbO and HbR with the modified Beer–Lambert law. Ideally, the concentration changes reflect neuronal activity in the superficial cortical layers [2]. Portable fNIRS systems have been developed to facilitate brain activation measurements in realistic settings with relatively freely moving subjects, e.g., during driving [3]. However, motion artifacts (MAs) induced in less restricted settings, e.g., by the movement of the probes, are an essential challenge to overcome [4].

MAs come in various forms, including shape and magnitude changes, as well as oscillations. MAs alter the form of the observed shape of the HbO and HbR signals, resulting in erroneous estimates of brain activity [5]. Generally, sudden optode movements generate spike-like artifacts [6], whereas optode position changes generate baseline shifts [7], steps [8], oscillations [9], and slow drifts [10]. MAs are easier to detect when they have large magnitudes and contain high-frequency oscillations. However, small-magnitude and slow, low-frequency MAs can be hard to detect. This can limit the effectiveness of MA correction algorithms that include an MA detection step.

The most consequent approach to treating MAs once they are detected is to reject the contaminated data segment. However, this can lead to unacceptable data loss. In order to remove MAs without data loss, researchers have recently developed several MA correction approaches. Currently, a gold standard for effectively removing the MA effects is still lacking [11]. Popular methods make strong assumptions about the nature of the hemodynamic response and the MA. Some require additional MA detection algorithms, e.g., spline interpolation and Targeted Principal Component Analyses (tPCAs). Others, such as principal component analyses (PCAs), correlation-based signal improvement (CBSI) [12], and wavelet filtering [13] can be run without MA detection. In addition, smoothing methods have been used to filter out MAs, e.g., Savitzky–Golay (SG) filtering [14] and robust locally estimated scatterplot smoothing (RLOESS) [15]. However, hybrid or combined methods, such as spline interpolation with a Savitzky–Golay filter [10], wavelet filtering with spline interpolation [16], and wavelet filtering with CBSI (WCBSI) [17], have proven to be more effective in MA correction than each method alone. It should be noted that all these methods can either leave residual MAs to various degrees or introduce new artifacts if their assumptions are not met or when the wrong parameters are chosen [17].

Some recent MA correction methods linearly combine direct movement measurements from accelerometers and gyroscopes with fNIRS to estimate and/or remove MAs from fNIRS data. Examples include an auto-regressive model with an exogenous input (ARX) [18], accelerometer data with canonical correlation analysis [11], and accelerometer-based motion artifact removal (ABAMAR), which uses the standard deviation of the accelerometer data to detect MAs in fNIRS and corrects baseline shifts [19]. Linear methods assume a linear relationship between the IMU measurements and the fNIRS artifact. In this study, we investigated the capabilities of the linear and nonlinear Hammerstein–Wiener model (HWM) to improve MA correction. Our HWM approach integrates fNIRS data with concurrent measurements from a three-axis accelerometer and gyroscope to estimate MAs in the fNIRS signal and uses the estimates to correct them. In addition, we applied a bandpass filter (BPF) to refine the data further. This filter enforces constraints on the signal bandwidth imposed by the slow hemodynamic response function (i.e., the system’s impulse response) and eliminates high-frequency noise as well as physiological artifacts from noise sources other than MAs [2,20]. 

We evaluated the performance of our suggested nonlinear MA correction method on empirical data and compared it to eight popular methods. In our experiment, we recorded brain activity in the motor cortex with fNIRS while participants performed a hand tapping task in three experimental conditions. In the first condition, participants kept their heads motionless while performing the task. This provided a ground truth signal of the expected brain activation. The second and third conditions required participants to perform small and large head movements while performing the task. We used this dataset to thoroughly assess the quantitative and qualitative aspects of our suggested approach and to validate its performance by comparing it with the most popular MA correction algorithms. For evaluation, we used four metrics that capture different aspects of algorithm performance: area under the curve (AUC), Root Mean Square Error (RMSE), Pearson correlation coefficient (R), and signal-to-noise ratio (SNR). A paired *t*-test was performed on all metrics to follow up on the significant differences in the performance of each tested algorithm. The dataset was made open access for future development and evaluations at https://www.doi.org/10.17605/OSF.IO/U3F89 (accessed on 12 May 2024).

## 2. Materials and Methods

### 2.1. Characteristics of the fNIRS Signal

The basic features of the interaction of near-infrared light (NIR) with human tissues relevant for functional neuroimaging are the following: (1) Human tissues are relatively transparent to light in the near-infrared spectral range (650–1000 nm) [21]. (2) NIR light is either absorbed by pigmented compounds (chromophores) or scattered in tissue [21]. (3) The dominant transport factor of NIR light in tissue is scattering, which is typically about 100 times more likely than absorption [22]. (4) Hemoglobin, which is located in vessels of the microcirculation, such as capillaries, arterioles, and venular beds, is a main contributor to the absorption of NIR light in the brain. The arterial blood volume fraction in the human brain is approximately 30% [23]. However, in fNIRS light, sources and detectors are attached to the scalp, and therefore the raw fNIRS signal is caused by a complex mixture of contributions by the blood supply and oxygen consumption in the brain, absorption in the tissues between sources and detectors (the scalp, bones, and the brain), and autonomous and heartbeat-related blood pressure changes, to name some [24]. Some of these non-brain influences can be mitigated by filtering techniques. Importantly, our study was designed to keep such factors comparable across experimental conditions.

### 2.2. Modified Beer–Lambert Law in the fNIRS Measurement

The fNIRS technique measures relative changes in light attenuation and uses a modified Beer–Lambert law to quantify changes in hemoglobin concentration as given by the following equation [25,26]:$$\Delta OD_{\lambda }=\frac{\Delta c}{\varepsilon_{\lambda }\;\cdot L\;\cdot DPF}$$
where $\Delta OD_{\lambda }$ is the optical density change, $\varepsilon_{\lambda }$ is the chromophore’s extinction coefficient, Δ*c* is the concentration change, *L* is the distance between the light entry and exit points, λ is the wavelength, and *DPF* is the differential pathlength factor. Most of the fNIRS devices use two wavelengths between 650 and 1000 nm to capture the HbO and HbR concentration changes [26,27].

### 2.3. Hammerstein–Wiener Model Configuration

The Hammerstein–Wiener model has its application in nonlinear system identification and integrates elements from both the Hammerstein and Wiener models. The Hammerstein model was first presented by Narendra and Gallman (1966) and the Wiener model by Norbert Wiener (1942). The Hammerstein–Wiener model applies a static nonlinearity to its input followed by a linear block. Conversely, the Wiener model passes the input through a linear block followed by a static nonlinearity. The Hammerstein–Wiener model combines both, resulting in a nonlinear–linear–nonlinear model structure as shown below in Figure 1. The Hammerstein–Wiener model represents a nonlinear system that is easier to implement and estimate than other linear or nonlinear models [28]. 

As depicted in Figure 1, we employed the Hammerstein–Wiener model with multiple inputs (IMU signals) and a single output (fNIRS signal). We used MATLAB’s System Identification Toolbox (MATLAB version R2020a) to estimate the nonlinear and linear parameters. The estimation report provides details of the model configuration, e.g., the names of regressors used in both (first and last) nonlinear blocks, along with suggestions for parameters. Based on this report, we selected a sigmoid network with ten units for the nonlinear block and a state space model for the linear block, which is configured as a transfer function with three poles and two zeros with a short input delay of one.

### 2.4. Hammerstein–Wiener Model (HWM) for Artifact Removal 

The whole MA-estimation-and-removal process with the HWM is outlined in Figure 1. We assume that the motion-contaminated fNIRS signal *z*(*t*) measured by the fNIRS optodes is the sum of the uncontaminated ground truth fNIRS signal *s*(*t*) and the MAs produced by head motions in the fNIRS signal *y*(*t*). This can be written as the following equation: (1)$$z\left(t\right)=s\left(t\right)+y\left(t\right)$$

Here, we used the HWM to estimate the MA in the fNIRS-measured signal. HW modelling assumes that the input–output relationship of a system can be decomposed into two or more interconnected elements when their output is nonlinear. More precisely, the HWM configuration consists of a static nonlinear block *m*1(.) cascading into a linear dynamic block *h*(τ), followed by another static nonlinear block *m*2(.) [29]. 

The first nonlinear block *m*1(.) transforms the input IMU data *u*(*t*) as the following: (2)$$x\left(t\right)=c_{1}\left(u\left(t\right)\right)=\sum_{i=1}^{m}\sum_{q1=0}^{Q1}c_{i}^{q1}u_{i}^{q1}\left(t-\tau \right),$$
where *u*(*t*) is the input with six dimensions (ax, ay, az, gx, gy, and gz) and *x*(*t*) is the one-dimensional output of the first nonlinear block. *i* is the counter and *m* is the number of IMU data channels, τ is the time delay, and *t* is the time interval. Cascading (2) into the linear stage, we have the following:(3)$$w\left(t\right)=\sum_{\tau =0}^{T-1}h\left(\tau \right)\left\{\sum_{i=1}^{m}\sum_{q1=0}^{Q1}c_{i}^{q1}u_{i}^{q1}\left(t-\tau \right)\right\}$$

Here, *T* is the memory length. Cascading the output from the linear stage into the second nonlinearity, and rearranging, we obtain the output equation for the Hammerstein–Wiener model MA estimation:(4)$$\hat{y}\left(t\right)=\sum_{i=1}^{m}\left\{\sum_{q2=0}^{Q2}c_{i}^{q2}{\left(\sum_{q1=0}^{Q1}c_{i}^{q1}\sum_{\tau =0}^{T-1}h\left(\tau \right)u_{i}^{q1}\left(t-\tau \right)\right)}^{q2}\right\}$$
where *Q*1 is the order of the first nonlinear kernel in Equation (2) and *Q*2 is the order of the second nonlinear kernel in Equation (4).

In order to calculate the corrected fNIRS signal $\hat{s}\left(t\right)$, we subtract the HWM MA-estimated signal $\hat{y}\left(t\right)\;$from the motion contaminated fNIRS signal *z*(*t*), as described by the following equation:(5)$$\hat{s}\left(t\right)=z\left(t\right)-\hat{y}\left(t\right)$$

### 2.5. Motion Artifact Correction Algorithms

Here, we briefly review the popular MA correction methods employed in this study for comparison with the HW method. We refer the reader to [30] for details of the methods.

Principal component analysis (PCA) finds an orthogonal transformation (principal components) to transform the original time series of N fNIRS channels into N uncorrelated time series of the principal components. If the magnitude of the MAs is larger than the brain signal and is correlated across sensors, then PCA may be able to separate MAs from brain signals [30]. Unfortunately, PCA applied to the entire measured signal often has difficulty distinguishing between brain and MA signals and may, therefore, partly remove the signal as well [31]. An extension of traditional PCA called Targeted Principal Component Analysis (tPCA) aims to improve MA and brain signal separation. This algorithm first detects MAs, then concatenates the MA-contaminated segments to estimate the PCA, and finally removes MAs only from those segments. The method has some complexity, as the user must set several parameters [32]. 

Correlation-based signal improvement (CBSI) is based on the hypothesis that HbO and HbR are always negatively correlated. MAs change this correlation because they have similar effects on both signals. Therefore, HbO and HbR are positively correlated in MA segments [12]. The MA correction is performed by estimating the parameters of a simple linear function that predicts, e.g., HbR from HbO data. However, a consequence of CBSI correction is that HbR and HbO become redundant, as one is predicted from the other. This may not be true in the actually acquired data [30]. 

Wavelet correction (WC) is a time–frequency method. It is separately applied to the time series of every fNIRS channel and decomposes the signal in a set of frequency bands with varying temporal resolutions. WC produces detail and approximation coefficients for every frequency resolution. The outliers of the distribution of the detail coefficients are associated with MAs. Therefore, MAs can be eliminated by setting these outlier coefficients to zero before performing the inverse wavelet transform to reconstruct the MA-corrected signal time series [13]. The other approaches employed here perform MA correction by local function fitting: The spline interpolation (SI) technique first applies an MA detection algorithm and fits a cubic spline to the MA intervals. Then, the interpolation is removed from the original signal to obtain the MA-corrected signal [33]. Savitzky–Golay (SG) filtering is related, as it locally fits a polynomial function to the MA interval and replaces the observed time series with the estimate from the fitted polynomial [14]. Moreover, robust locally estimated scatterplot smoothing (RLOESS) first fits a locally weighted regression, which is then iteratively adjusted depending on the magnitude of the residuals [15].

WCBSI combines the wavelet with the CBSI approach for MA correction. In a previous study [17], WCBSI was proven to be the most effective MA correction method among the methods reviewed in this section across all quality measures reported in the current study.

### 2.6. Participants 

We originally recorded data from twenty participants, but we lost the IMU data from three of them, so the analysis is based on seventeen healthy participants with an average age of 29 ± 4 years, including twelve females. Ten had various shades of brown hair, six had black hair, and one had red hair. Individual participants’ age, gender, and hair color are listed in Appendix A. Participants had no neurological or psychological disorders. We obtained their written informed consent prior to their participation in the study. The IRB of Carl von Ossietzky Universität Oldenburg approved the study protocol under the code (Drs. EK/2020/021-01). 

### 2.7. Experimental Procedure and Study Design

We measured the fNIRS, accelerometer, and gyroscope data while participants performed a hand tapping task and concurrent head movements with different amplitudes. A condition without head movements allowed us to empirically determine the ground truth fNIRS signal elicited by hand tapping. This approach allowed us to have significant control over the experiment. The experimental procedure was the same as in [17,34]. 

The experiment was performed in a dark and soundproof room. The participants were seated comfortably in front of a 24-inch LCD computer monitor at an approximately 75 cm distance. We used Psychtoolbox (v3.0.10), a MATLAB toolbox [35], to implement three tasks. Each trial involved 10 s of activity followed by a 20 s rest period for each task (Figure 2). This cycle was repeated 25 times, with a fixed task sequence, resulting in a total of 75 trials with a 41.5 min total data acquisition time per participant [17]. The experiment started and ended with a 120 s long baseline measurement without any overt activity from the participants. Psychtoolbox sent two signals at the beginning of each task to synchronize the IMU and the fNIRS recordings. (1) The event time for each task was sent to the fNIRS recording software (Oxysoft v3.0.103.3) through the lab streaming layer. (2) Simultaneously, a beep sound was sent to the Arduino board through the auxiliary port and was recorded in an extra channel in parallel with the IMU recordings.

The participants were instructed to initially place their hands on the table and to avoid head movements. Upon the appearance of the word ‘tapping’ on the screen, they started tapping their hand on the table, keeping their head steady in the first task (T1). This task served to record the ground truth signal. In the second task (T2), participants were asked to move their heads slightly while tapping their hands. The third task (T3) was similar to the second but with more substantial head movements. The head movements included flexion–extension (forward–backward) and left–right movements performed in random order. The hand tapping was performed at approximately 1 ± 0.5 Hz [17,36]. Participants practiced the tasks for 5–10 min before the experiment started to ensure that they followed the procedure correctly. 

### 2.8. fNIRS and IMU Data Collection and Processing 

We employed a wearable fNIRS Artinis medical OctaMon system to measure experimentally induced HbO and HbR changes in the motor cortex. The system consisted of eight sources emitting light at 850 and 760 nm wavelengths and two detectors sampled at 10 Hz. Sources and detectors were integrated into a specialized cap, with holder positions following the international transcranial positioning pattern 10/20 [37]. Detectors were placed in positions C3/C4 and sources in positions C1/C2, C5/C6, FC3/FC4, and CP3/CP4 (Figure 3). This created four fNIRS channels approximately placed over the hand area of the primary somatomotor cortex of each hemisphere. Before data collection, we checked the signal quality using the Oxysoft v3.0.103.3 recording program. This assessment included checking the light intensity and potential ambient light interference (˂1% of the total signal) and ensuring that the light wavelengths emitted from each source fell within the specified range of 760 ± 5 nm and 850 ± 6 nm. 

We used two MPU6050 IMUs (each housing a 3-axis accelerometer and a gyroscope; AZ Delivery Vertriebes GmbH, Deggendorf, Germany) to measure linear and rotational movement components. Both were firmly fixed with custom-made holders for precise measurements of the detector and head movements. One IMU was fixed centrally at the position FCZ on the cap (head-IMU). The other IMU was fixed to the top of the fNIRS detector used in the analysis, which was at position C4 (probe-IMU) (see Figure 3). Both IMU sensors were recorded at a sampling frequency of 10 Hz, the same as the fNIRS sampling frequency, through an Arduino UNO board (for the code, see Appendix A), and the data were stored on a PC. 

### 2.9. Data Processing 

Before the data recording, we checked the signal quality of all tested channels according to the user guide of the fNIRS (Artinis medical system) using the manufacturer’s software (Oxysoft v3.0.103.3). The data processing was performed using the Homer3 v1.31.2 NIRS processing MATLAB toolbox [2], which implements the popular MA correction methods employed here. For HWM-based MA correction, we used the System Identification ToolboxTM [38] in MATLAB version R2020a. The block diagram in Figure 4 depicts the processing steps for all MA correction methods. 

The first step is applying the modified Beer–Lambert law to convert the raw intensity fNIRS measurements to the optical density (OD) data. The value of the differential pathlength factor depends on the age of the participant, and was determined by the method of Scholkmann and Wolf [39], which is integrated into the fNIRS manufacturer’s software (Oxysoft v3.0.103.3), and this factor averaged to 6 ± 0.5 for 760 nm and 850 nm wavelengths [40]. 

The HWM estimation is applied separately with the head-IMU and the probe-IMU data. The accelerometer and gyroscope data are then run through the estimated HWMs to estimate the MA, which is subtracted from the fNIRS signal, resulting in the HWM-corrected fNIRS signals. HWMH refers to head-IMU correction and HWMP to probe-IMU correction. 

A 3rd-order Butterworth band-pass filter with a cut-off frequency of 0.01–0.1 Hz is applied to the corrected data to reduce very slow drifts and high-frequency noise [2,20]. Finally, to estimate the mean HbO and HbR response elicited by the hand tapping, all epochs in each experimental task (T1, T2, and T3) were averaged over a 20 s long interval, starting with the hand tapping. This results in three mean HbO and three mean HbR responses (a ground truth hand tapping response signal, a hand tapping with a small-head-movement response signal, and a hand tapping with a large-head-movement response signal). 

For the correction techniques that require defined artifact intervals (spline, splineSG, and tPCA), we used amplitude and standard deviation thresholds to mark the data samples with MAs. All parameters for motion detection and the MA correction algorithms were the same as in [2]. 

### 2.10. Metrics of Comparison 

In order to quantify the performance of the MA correction approaches, we compared the averaged 20 s epochs of the $\hat{HRF}$ motion-corrected signal to the average HRF of the ground truth signal obtained without head movements individually for the HbO and HbR signals at different levels of motion artifacts (T2 and T1), and for each technique. We used four metrics for this comparison: 

The signal-to-noise ratio (SNR), measured in decibels, relates the ground truth-signal power to the power of the residual noise in the motion-corrected signal. It is calculated as in [5], where the ground truth signal is *HRF*(*i*) with *i* = 1, …. N, the sample index. $\hat{HRF}$(i) is the motion-corrected signal. By subtracting both, we can calculate the residual noise signal *E*(*i*).
(6)$$E\left(i\right)=\hat{HRF}\left(i\right)-HRF\left(i\right),\;$$
and with that, the SNR
(7)$$\mathrm{SNR}=10*log_{10}\left(\frac{\sum_{i=1}^{N}\;{\left(HRF\left(i\right)\right)}^{2}}{\sum_{i=1}^{N}{\left(E\left(i\right)\right)}^{2}}\right)$$

The Area under the curve difference (Δ*AUC*) is a global measure that compares the overall deviation of two curves (ground truth and motion-corrected signal) from the baseline. We calculated $AUC_{HRF\left(i\right)}$ and $AUC_{\hat{HRF}\left(i\right)}$ using a MATLAB function for numerical integration (trapz). The difference is calculated using the following formula:(8)$$\Delta AUC=AUC_{HRF\left(i\right)}-AUC_{\hat{HRF}\left(i\right)}$$

The Root Mean Square Error (RMSE) measures the average deviation between the ground truth signal and the motion-corrected signal using the following equation:(9)$$\mathrm{RMSE}=\sqrt{\frac{\sum_{i=1}^{N}{\left(HRF\left(i\right)-\hat{\;HRF}\left(i\right)\right)}^{2}}{N}}$$

Here, N is the number of samples in an epoch and *i* is the sample count. Note that RMSE is sensitive to shape and scaling differences. 

Pearson’s correlation coefficient (R) measures the similarity of the shape of the ground truth $HRF\left(i\right)$ and the motion-corrected signals $\hat{HRF}\left(i\right)$. The Pearson correlation ranges between −1 and 1 and is insensitive to scaling differences between the two signals.

### 2.11. Statistical Analysis 

In order to demonstrate that the HWM can reduce MAs, we compared the quality metrics (SNR, Δ*AUC*, RMSE, and R) between the HWM motion-corrected signal and the uncorrected signal with paired *t*-tests. For the comparison with the other MA corrections, we built on prior knowledge from our previous study [17], in which we demonstrated that WCBSI has the best MA correction performance among all comparison methods and quality measures employed in the present study. Therefore, in order to avoid an excessive number of statistical tests, we only statistically compared HWM to WCBSI, the “best in class” in the previous study. For each quality metric, we added a Benjamini–Hochberg correction with a false discovery rate of 5%. In addition, we calculated Cohen’s d as a measure of the effect sizes. All statistics were performed using IBM SPSS (v29.0.0.0). 

### 2.12. Measuring Participants’ Head Orientation during Experimental Tasks 

We measured participants’ head motion in the experimental tasks (T1, T2, and T3) by calculating the rotational angles of the head-IMU accelerometer data (X, Y, and Z). Figure 2 shows pitch (*θ*) for the *X*-axis, roll (*ψ*) for the *Y*-axis, and yaw (*φ*) for the *Z*-axis. These angles are expressed according to the below formula in degrees relative to the Earth’s gravity g = 9.81 m/s2 [41].
$$\theta =arctan\left[\frac{a_{x}}{\sqrt{a_{y}^{2}+a_{z}^{2}}}\right]\times \frac{180}{\pi },\;\mathrm{is}\;\mathrm{the}\;\mathrm{rotation}\;\mathrm{angle}\;\mathrm{of}\;\mathrm{the}\;x\text{-axis}.$$
$$\psi =arctan\left[\frac{a_{y}}{\sqrt{a_{x}^{2}+a_{z}^{2}}}\right]\times \frac{180}{\pi },\;\mathrm{is}\;\mathrm{the}\;\mathrm{rotation}\;\mathrm{angle}\;\mathrm{of}\;\mathrm{the}\;y\text{-axis}.$$
$$\varphi =arctan\left[\frac{a_{z}}{\sqrt{a_{x}^{2}+a_{y}^{2}}}\right]\times \frac{180}{\pi },\;\mathrm{is}\;\mathrm{the}\;\mathrm{rotation}\;\mathrm{angle}\;\mathrm{of}\;\mathrm{the}\;z\text{-axis}.$$
where *a_x_*, *a_y_*, and *a_z_* are the acceleration output on the (*x*-, *y*-, and *z*-) axis. 

### 2.13. Head- and Probe-IMU Sensors’ Relationship Based on Canonical Correlation Analysis (CCA)

Canonical correlation analysis (CCA) was first introduced by Hotelling (1936) with the goal of fitting projections of two datasets into a common space in which they are maximally correlated. Here, we use CCA to align the six-dimensional recording of the two IMU sensors (head-IMU and probe-IMU) in a common space to analyze their relationship. 

## 3. Results 

### 3.1. Qualitative Comparison of Motion Artifact Correction Methods 

In this section, we provide a brief qualitative comparison of how different MA correction methods cope with different types of MAs. Figure 5 provides an illustrative example of single-trial HbO and HbR signals with various MA types marked in red. These include baseline shifts, drifts, and spikes. It can be seen in these examples that for both small (T2) and large (T3) head movements, the nonlinear HWM approach excels in eliminating the high-amplitude spikes, rectifying the up- and down-baseline shifts, and successfully mitigating slow drifts. All methods seem to correct both up and down spikes, but this may be at least partly due to the low-pass filters applied as the last step after MA correction. Of the other MA correction methods tested, only HWMH, HWMP, WCBSI, and RLOESS corrected the downshift and slow drift in T2. Furthermore, in T3, our suggested HWMH, HWMP, and WCBSI succeeded in remedying the up-baseline shift, but none of the other techniques did. Notably, HWMH and HWMP perform best in resetting the offset amplitude of the signal in all three tasks. Moreover, only HWMH and HWMP maintain consistent fNIRS response amplitudes across all three tasks (T1, T2, and T3). Note how well the HWM-corrected HbO and HbR time courses follow the expected time courses (positive for HbO and negative for HbR), with comparable peak amplitudes toward the end of the movement interval despite large head movements. Also, note the similarity between the HWM corrections obtained with the head- and the probe-IMUs. In the following, we will provide a quantitative evaluation of the performance of all MA correction algorithms.

### 3.2. Quantitative Comparison of Motion Correction Methods

All measures of MA correction quality are referenced to the average tapping-related fNIRS activations measured in the T1 epochs without head movement. Consequently, no MA correction was applied to obtain T1 averages. We consider them to be the empirical ground truth to which we compare the MA-contaminated and the MA-corrected fNIRS responses measured in the T2 and T3 epochs with small and large head movements, respectively. As depicted in Figure 4, averages for comparison with the ground truth signal were taken after MA correction. To quantify the results, we calculated four distinct quality metrics, capturing different aspects of signal quality (see the Methods Section): the SNR capturing the residual noise level, ΔAUC, the RMSE capturing the magnitude and shape reconstruction combined, and the Pearson correlation coefficient R capturing shape reconstruction alone. The epochs to calculate the metrics were 20 s long and started with the beginning of the respective tasks. 

Our main interest was in the comparison of the performance of our novel nonlinear HWM MA correction approach with the uncorrected average epochs (i.e., averaging as the only MA treatment) and the linear WCBSI correction that we recently suggested [17] and proved to be superior to all alternative MA correction methods tested in the previous study (spline, spline SG, RLOESS, CBSI, PCA, tPCA, and wavelet) and here. Therefore, in order to reduce the number of statistical comparisons among MA correction methods, we focused on testing differences between the HWM- and WCBSI-corrected signals. Furthermore, our results indicated that MA correction success is virtually identical with the head-IMU (HWMH) and the probe-IMU (HWMP). Therefore, we averaged the quality measures over the two corrections into a single value for the HWM. We used this average value for the statistical comparisons.

All tests were conducted as two-tailed paired *t*-tests. In the plots below, we use the symbol (*) to indicate significance levels *p* < 0.05 (**) for *p* < 0.01 and (***) for *p* < 0.001 [11]. We added Benjamini–Hochberg-corrected *p*-values for a false discovery rate (FDR) of 5% with *n* = 8 comparisons (two signals (HbO and HbR), two movement levels (T2 and T3), and two methods (WCBSI and HWM) for each quality measure and calculated Cohen’s d as a measure of the effect size.

Signal-to-noise ratio (SNR) relates the power of residual noise after MA correction to the power in the ground truth signals. A higher SNR value indicates a better MA correction. Figure 6 summarizes the averaged SNR (17 participants) for the HbO and HbR signals after MA correction in tasks T2 and T3 for each correction approach. The SNRs differ very little between HWM types (all two-tailed paired *t*-tests with df = 16: HbO T2: *p* > 0.05, Cohen’s d = 0.451; HbO T3: *p* > 0.05, Cohen’s d = 0.287, HbR T2: *p* > 0.05, Cohen’s d = 0.448; HbR T3: *p* > 0.05, Cohen’s d = 0.262). Therefore, we averaged the HWMP and HWMH SNRs for HWM correction in the following analysis. 

Compared to uncorrected signals, the SNR increased significantly with HWM correction for both HbO and HbR in T2 and T3 (all two-tailed paired *t*-tests with df = 16: HbO T2: *p* < 0.001, Cohen’s d = 1.918; HbO T3: *p* < 0.001, Cohen’s d = 3.620; HbR T2: *p* < 0.001, Cohen’s d = 2.012; HbR T3: *p* < 0.001, Cohen’s d = 2.180). All the comparisons are still significant after Benjamini–Hochberg correction for n = 8 comparisons at an FDR of *p* < 0.05 (corrected *p*-values: HbO T2: *p* < 0.001; HbO T3: *p* < 0.001; HbR T2: *p* < 0.001; HbR T3: *p* < 0.001). 

Compared to WCBSI, the SNR increased numerically with HWM correction for both HbO and HbR in T2 and T3 (see Figure 6). This difference reached significance in the uncorrected paired *t*-tests (all two-tailed paired *t*-tests with df = 16: HbO T2: *p* < 0.05, Cohen’s d = 0.530; HbO T3: *p* < 0.001, Cohen’s d = 1.46; HbR T2: *p* < 0.05, Cohen’s d = 0.692; HbR T3: *p* < 0.05, Cohen’s d = 0.700). Also, after Benjamini–Hochberg correction for n = 8 comparisons at an FDR of *p* < 0.05, the measures were significantly different (corrected *p*-values: HbO T2: *p* < 0.05; HbO T3: *p* < 0.001; HbR T2: *p* < 0.05; HbR T3: *p* < 0.05).

In sum, we obtained for both head movement levels the highest SNRs with the HWM correction. The SNR reached up to 13–14 dB for both HWMH and HWMP, which amounts to a ratio of ca 20:1 to 25:1. 

The area under the curve difference (ΔAUC) measures the difference between the integrals of the ground truth signal and the MA-corrected signals. This is a global measure that can, for example, detect signal rescaling, and smaller values indicate better MA correction. Reported alone, it can be hard to interpret. Figure 7 shows the ΔAUCs for the different correction approaches for HbO and HbR separately. Note that the HbR signals were multiplied by −1 to compensate for the sign flip. The ΔAUC did not show significant differences between HWMP and HWMH correction (all two-tailed paired *t*-tests with df = 16: HbO T2: *p* > 0.05, Cohen’s d = 0.316; HbO T3: *p* > 0.05, Cohen’s d = 0.041; HbR T2: *p* > 0.05, Cohen’s d = 0.139; HbR T3: *p* > 0.05, Cohen’s d = 0.141). Therefore, we averaged the HWMP and HWMH ΔAUCs for HWM correction in the following analysis. 

Compared to uncorrected signals, ΔAUC decreased significantly with HWM correction for both HbO and HbR in T2 and T3 (all two-tailed paired *t*-tests with df = 16: HbO T2: *p* < 0.001, Cohen’s d = 1.011; HbO T3: *p* < 0.001, Cohen’s d = 1.195; HbR T2: *p* < 0.01, Cohen’s d = 0.730; HbR T3: *p* < 0.001, Cohen’s d = 2.180). All effects remained statistically significant after Benjamini–Hochberg correction for n = 8 comparisons at an FDR of *p* < 0.05 (corrected *p*-values: HbO T2: *p* < 0.05, HbR T2: *p* < 0.05, HbO T3: *p* < 0.001, HbR T3: *p* < 0.01). 

Compared to WCBSI, ΔAUC decreased numerically with HWM correction for both HbO and HbR in T2 and T3 (see Figure 7). This difference reached significance in the uncorrected paired *t*-tests (all two-tailed paired *t*-tests with df = 16: HbO T2: *p* < 0.001, Cohen’s d = 1.120; HbO T3: *p* < 0.001, Cohen’s d = 1.081; HbR T2: *p* < 0.01, Cohen’s d = 0.708; HbR T3: *p* < 0.001, Cohen’s d = 0.964), After Benjamini–Hochberg correction for n = 8 comparisons at an FDR of *p* < 0.05, all effects remained significant (corrected *p*-values: HbO T2: *p* < 0.05; HbO T3: *p* < 0.05; HbR T3: *p* < 0.05; HbR T2: *p* < 0.05).

In sum, similar to SNR, the HWM produced for ΔAUC the best MA correction results over the range of head movements investigated (T2 ΔAUC: 5.2 × 10^−5^ for HbO and 3.5 × 10^−5^ for HbR; T3 ΔAUC: 3.7 × 10^−5^ for HbO and 4.6 × 10^−5^ for HbR). 

The Root Mean Square Error (RMSE) quantifies the unscaled averaged absolute discrepancy between the ground truth signal and the MA correction signals. Lower RMSE scores signify more effective MA correction. Figure 8 depicts the RMSE measures of MA correction success in the experimental conditions T2 and T3. There are only minor RMSE differences between HWMP and HWMH (all two-tailed paired *t*-tests with df = 16: HbO T2: *p* > 0.05, Cohen’s d = 0.361; HbO T3: *p* > 0.05, Cohen’s d = 0.267; HbR T2: *p* > 0.05, Cohen’s d = 0.538; HbR T3: *p* > 0.05, Cohen’s d = 0.423). Therefore, we calculated the following analysis using the average HWM-correction RMSE score.

Compared to uncorrected signals, the RMSE decreased for both HbO and HbR in T2 and T3 (all two-tailed paired *t*-tests with df = 16: HbO T2: *p* < 0.05, Cohen’s d = 0.494; HbO T3: *p* < 0.05, Cohen’s d = 0.624; HbR T2: *p* < 0.05, Cohen’s d = 0.611; HbR T3: *p* < 0.01, Cohen’s d = 0.903). Most of the tests remained significant after Benjamini–Hochberg correction for n = 8 comparisons at an FDR of *p* < 0.05; the comparison measures were significant (corrected *p*-values: T2; HbO T2: *p* < 0.05, HbR T2: *p* < 0.05, and HbR T3: *p* < 0.01). Only the comparison of the HbO RMSE scores for the large head movements (T3) did not pass correction (HbO T3: *p* > 0.05). We consider the HWM-correction effect to be statistically marginal but note that the mean numerical RMSE was smaller for all HWM-corrected than for uncorrected signals (see Figure 8). 

Compared to WCBSI, the RMSE decreased numerically with the HWM correction for both HbO and HbR in T2 and T3 (see Figure 8). This difference reached significance in the uncorrected paired *t*-tests only in condition T2, the smaller head movement (all two-tailed paired *t*-tests with df = 16: HbO T2: *p* < 0.05, Cohen’s d = 0.593; HbO T3: *p* > 0.05, Cohen’s d = 0.421; HbR T2: *p* < 0.01, Cohen’s d = 0.870; HbR T3: *p* > 0.05, Cohen’s d = 0.470). The same pattern emerges after Benjamini–Hochberg correction for n = 8 comparisons at an FDR of *p* < 0.05: only the differences for T2 remained significant (corrected *p*-values: HbO T2: *p* < 0.05; HbO T3: *p* > 0.05; HbR T2: *p* < 0.01; HbR T3: *p* > 0.05).

In sum, we obtained consistently the best numerical RMSE scores with the HWM correction. However, the differences between the uncorrected and WBCSI-corrected RMSEs tended to be marginal in the larger-head-movement condition T3 and did not survive correction for multiple comparisons. 

Pearson’s correlation coefficient (R) captures the relative shape similarity between the ground truth signal and the MA-corrected signals. R is a normalized value, where a value of one indicates that the shapes of the two curves are perfect reproductions of each other, and zero indicates that they are unrelated. Note that R is insensitive to scaling differences. Figure 9 shows the correlation coefficients for the different correction approaches for HbO and HbR separately. The highest correlations are obtained with the proposed HWMH and HWMP approaches. Again, the correlation coefficients obtained with HWMP and HWMH are very similar and do not differ significantly (all two-tailed paired *t*-tests with df = 16: HbO T2: *p* > 0.05, Cohen’s d = 0.113; HbO T3: *p* > 0.05, Cohen’s d = 0.402; HbR T2: *p* > 0.05, Cohen’s d = 0.085; HbR T3: *p* > 0.05, Cohen’s d = 0.329). Therefore, we calculated the following analysis with the averaged HWM Pearson correlation coefficient in the following analysis. 

Compared to uncorrected signals, R increased significantly for both HbO and HbR in T2 and T3 (all two-tailed paired *t*-tests with df = 16: HbO T2: *p* < 0.001, Cohen’s d = 1.498; HbO T3: *p* < 0.001, Cohen’s d = 0.941; HbR T2: *p* < 0.001, Cohen’s d = 1.329; HbR T3: *p* < 0.001, Cohen’s d = 1.696). Also, after Benjamini–Hochberg correction for n = 8 comparisons at an FDR of *p* < 0.05, all comparisons remained significant (corrected *p*-values: HbO T2: *p* < 0.001; HbO T3: *p* < 0.01; HbR T2: *p* < 0.001; HbR T3: *p* < 0.001).

HWM correction achieves similarly accurate shape reconstructions to WCBSI (see Figure 9). The mean R overall conditions are for HWM 0.86 (STD = 0.03) and for WCBSI 0.86 (STD = 0.02). Consequently, there were no significant differences between any conditions (all two-tailed paired *t*-tests with df = 16: HbO T2: *p* > 0.05, Cohen’s d = 0.226; HbO T3: *p* > 0.05, Cohen’s d = 0.190; HbR T2: *p* > 0.05, Cohen’s d = 0.054; HbR T3: *p* > 0.05, Cohen’s d = 0.271). Also, after Benjamini–Hochberg correction for n = 8 comparisons at an FDR of *p* < 0.05, no comparison was significant (corrected *p*-values: HbO T2: *p* > 0.05; HbO T3: *p* > 0.05; HbR T2: *p* > 0.05; HbR T3: *p* > 0.05). In addition, Pearson correlations were remarkably similar across head movement levels (T2 HbO: R = 0.89 vs. T3 HbO R = 0.86 and T2 HbR: R = 0.87 vs. T3 HbR R = 0.81). 

Given the observation that RMSE values were somewhat lower for the HWM than for WCBSI, and the fact that RMSE is sensitive to scaling and shape whereas Pearson correlation is only sensitive to shape, this suggests that the difference between WCBSI and HWM may be found mostly in scaling differences, at least at the smaller-head-movement levels in T2. Importantly, reliable response-amplitude estimates are key in many analysis approaches that operate on MA-corrected data.

### 3.3. IMU Data Analysis

#### 3.3.1. Head Orientation Analysis across Experimental Tasks

In this section, we analyze the head motion characteristics in the three experimental conditions. Figure 10 shows the mean pitch (θ), roll (ψ), and yaw (φ) head rotations for the three tasks, derived from the accelerometer data of the head-IMU. As instructed, head orientation remained stable during the T1 task, from which we derived the ground truth fNIRS responses. T2 and T3 show considerable head rotations, and those increased from T2 to T3 (T2: θ∼10°, ψ∼7°, and φ∼4°, T3: θ∼12°, ψ∼8°, and φ∼7°).

#### 3.3.2. Canonical Correlation Analysis (CCA) of the Head- and Probe-IMU Motion Signals

We originally assumed that the head-IMU and the probe-IMU might measure at least partly independent motion signals, as only the probe-IMU could measure the probe in addition to the head movements. In practice, it turned out that both IMUs combined with the HWM provide indistinguishable motion correction performance. This suggests that the simpler head mount was sufficient for the HWM correction in our recording setup. However, an equal level of MA correction does not necessarily mean that the two IMUs measured comparable motion acceleration signals, as the HWM may adapt to different inputs. Here, we wanted to see if the actual movement signals measured were comparable between the two IMU positions. As the IMUs are attached with different orientations, their sensor channels are not directly comparable. To align them, we used CCA, which finds a common space in which linear combinations of the original sensor channels (the CC factors) are maximally correlated. If both IMUs measured comparable motion signals, CCA should find a transformation for each sensor such that the CCA time series of the head- and probe-IMUs are highly correlated. Figure 11 depicts the mean CCA correlations for the two sensors. The three CCA channels for the accelerometers indicate a very high correlation, indicating that the IMU accelerometers measure virtually identical signals. For the gyroscopes, one CCA channel has a very high correlation and the second a somewhat lower one. The fact that the correlation in the third CCA channel is low is likely due to the nature of the head movements, which rotate around only two axes. Therefore, CCA can only find two correlated rotation axes. In sum, this analysis implies that head- and probe-IMUs measured very similar signals in our fNIRS measurement setup.

## 4. Discussion 

The main goal of this study was to develop and test a novel application of the linear–nonlinear Hammerstein–Wiener model for optimizing the MA correction of fNIRS measurements in a realistic experiment and to compare this approach with the most popular MA correction techniques. Notably, our approach provides the best, or at least on par with the “best in class”, MA correction, WCBSI, of the other investigated methods in several MA correction quality metrics, for both hemoglobin signals (HbO/HbR) and among a range of movement levels (small and large movement tasks). For both hemoglobin signals and movement tasks, the HWM correction method produces the best SNR of the MA corrected signals among all MA correction methods investigated. The HWM-corrected fNIRS also had the lowest values for differences in AUC and RMSE, indicating that it best recovers the shape and amplitude of the ground truth signal. This exceptional performance, reflected by the lowest value of (ΔAUC) among all other techniques, underscores its pronounced capability to effectively mitigate the high-amplitude spikes and high up–down-baseline shifts by successfully resetting the signal’s baseline offset. The lowest RMSE values for HWM-corrected fNIRS indicate minimal errors between the ground truth signal and MA-corrected signals compared to the other tested algorithms. Additionally, the HWM correction significantly improves the Pearson correlation between the MA corrected signal and the empirically measured ground truth signal compared to the uncorrected signal. Interestingly, the WCBSI method, which we previously developed [17], is on par with the HWM correction regarding the Pearson correlation. This suggests that WCBSI recovers the shape of the fNIRS signal with comparable quality to the HWM. However, the lower performance of WCBSI in the other quality measures indicates that it does not recover the amplitude of the signal as well as the HWM. Finally, visual inspection indicates optimal correction of segments with very low-amplitude and slow MAs by HWM, a task beyond the capability of other techniques.

There are two important differences between the HWM MA correction method developed here and the other popular approaches we tested. First, the HWM method uses IMU-based measurements of the head movements and integrates them into the MA correction process. This is different from the other MA correction approaches, which do not use movement information but try to correct MAs based on statistical signal features, employing signal processing techniques, or making assumptions about the relation between HbO and HbR signals. When these assumptions are violated, these techniques may fail to various degrees. Second, HWM correction is a nonlinear technique, while most other techniques are linear. The superior performance of the nonlinear HWM correction method indicates that the linearity of the MA is a strong assumption that is likely violated in practice. Different types of movement artifacts may not only have different dynamics, but also different amplitude scaling behavior, which is not possible to capture with linear methods. This may be the reason why the linear methods are good at correcting some types of MA but not others, as indicated by the examples in Figure 5. 

The HWM method was insensitive to differences in the placement of the IMUs used for correction in our fNIRS measurement setup, in which we used a standard cap with optode holders. This suggests that IMU placement can be solved in a relatively simple way by attaching it to the optode holder cap. An open question is whether the use and integration of multiple IMUs can further improve the HWM correction. In principle, the approach can handle inputs from multiple IMUs. However, the CCA analysis indicates that multiple IMUs may provide redundant information. In this case, no further improvement can be expected.

The advantage of the HWM method lies in its ability to estimate the MA based on IMU data by using the linear–nonlinear Hammerstein–Wiener model’s integration of the participants’ head movement data and the fNIRS data. This allows for precise quantification and removal across a wide range of movement levels. Another advantage of the HWM method is that it is a fully automated correction technique that does not require user-defined parameter entry for each movement level. Other MA correction methods rely on the manual selection of appropriate threshold parameters that must be adjusted based on the level of MA contamination before MA correction. Importantly, opposed to the “best in class” WCBSI, the HWM method corrects the HbR signal independently of the HbO signal. This way, it provides independent HbR and HbO estimates. Conversely, (W)CBSI combines the HbR and HbO signals in one synthetic signal that can be expressed as HbO or as HbR, but both synthetic signals are fully linearly dependent. Another important advantage is that the HWM method performs comparably over the movement levels employed here. This robustness against MA-level variations is consistent with [18,42]. 

A limitation of this study is that all processing was conducted offline. Although most fNIRS studies implement offline analyses in some use cases, online MA correction may be desirable. However, processing time may become an issue with larger sensor arrays because the HWM correction method is a channel-by-channel processing technique, and processing time scales approximately linearly with the number of fNIRS channels. Appendix A shows that the processing time for a single fNIRS channel consisting of HbO and HbR signals amounts to approximately 30 s for 40 min of data. This is comparable to most other techniques (16–690 s). However, for whole-head recordings with more than 100 fNIRS channels, the processing time may become considerable. Notably, the HWM correction was faster than that of wavelet, WCBSI, and RLOESS, which took 62, 65, and 690 s, respectively. However, future investigations should test the capability to perform online corrections. Another limitation of our study pertains to the performance of HWM correction in real-world recordings where head movements, e.g., during walking, may be even stronger or MAs have different dynamics. This requires further investigations into new datasets recorded under diverse conditions. However, the observation that the HWM can correct MAs caused by different levels of head movements and a range of different types of MAs without a loss in accuracy opens the possibility that it could also produce favorable results in less controlled fNIRS recording settings. Finally, the HWM correction requires additional head movement recordings, for example, from an IMU. These can be obtained with a commercial movement-tracking system or, as was done in this study, with a relatively cheap custom-made system. The MPU6050 three-axis accelerometer and gyroscope IMU used here are relatively cheap consumer-grade devices that were read at the low sampling rate of the fNIRS device with a cheap standard Arduino Uno microcontroller board. The code for the Arduino board is provided in the Appendix A. With that, it should be possible to reproduce our head motion measurement setup with minimal soldering skills. 

## 5. Conclusions

In the current study, we developed and tested a novel application of the nonlinear Hammerstein–Wiener model for combining the IMU-based head movement recordings with fNIRS recordings for the correction of fNIRS motion artifacts. According to four commonly used quality metrics (SNR, ΔAUC, RMSE, and R), the IMU-based HWM approach significantly improves MA removal compared to several popular MA correction techniques. Furthermore, we found that the placement of the IMU sensor on the participant’s head or on the top of the fNIRS probe did not significantly affect the performance of the HWM approach. Based on our results, we conclude that combining HWM with IMU-based motion data is the most favorable approach to improve the quality and reliability of the MA correction of fNIRS signals.

## Figures and Tables

**Figure 1 sensors-24-03173-f001:**
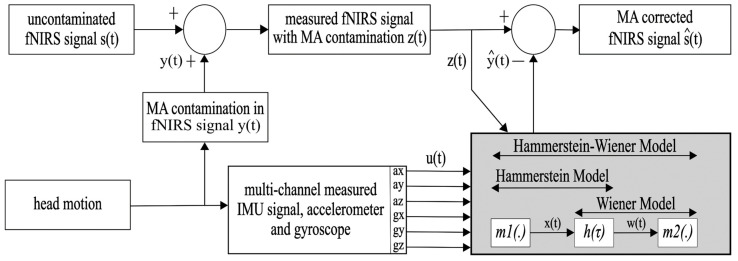
The block diagram of the HWM MA correction pipeline. Here, *s*(*t*) is the MA-uncontaminated fNIRS signal, and *y*(*t*) is the MA contamination by the head motion in the fNIRS signal. The measured fNIRS signal with the MA contamination is *z*(*t*), and *u*(*t*) holds the three-axis accelerometer and gyroscope signals. Both *z*(*t*) and *u*(*t*) are the inputs of the model. The HWM model (grey box) consists of two nonlinear blocks (m1 and m2) and one linear block *h*(τ). The model output $\hat{y}$(*t*) represents the estimated MA signal, which is subtracted from the measured fNIRS signal with MA contamination *z*(*t*) to obtain the $\hat{s}$(*t*), the MA-corrected fNIRS signal.

**Figure 2 sensors-24-03173-f002:**
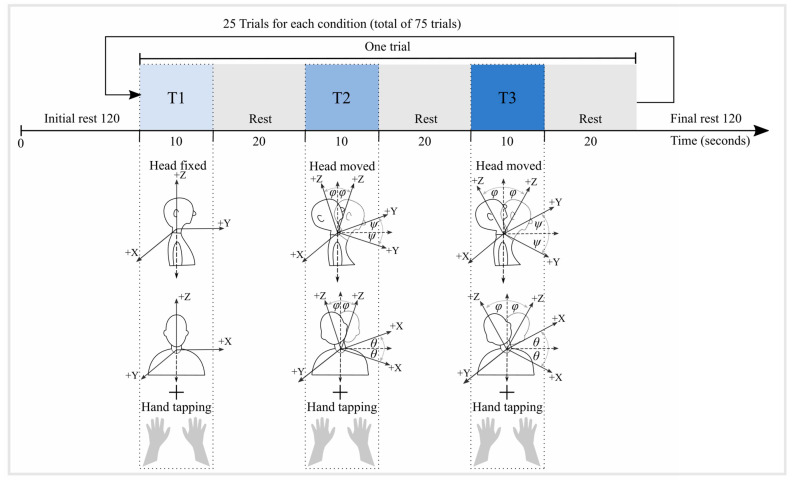
The experimental procedure and study design. The experiment begins with a 120 s initial rest period, followed by 10 s of task T1 (light blue box), involving hand tapping without head movements. After a 20 s rest period, 10 s of T2 (blue box) follows, involving hand tapping with small head movements (left–right–forward–backward). After another 20 s rest period, 10 s of task T3 follows (dark blue box), involving hand tapping and larger head movements. This sequence is repeated 25 times, resulting in 75 movement blocks (25 blocks for each task). The experiment concludes with a final 120 s rest period.

**Figure 3 sensors-24-03173-f003:**
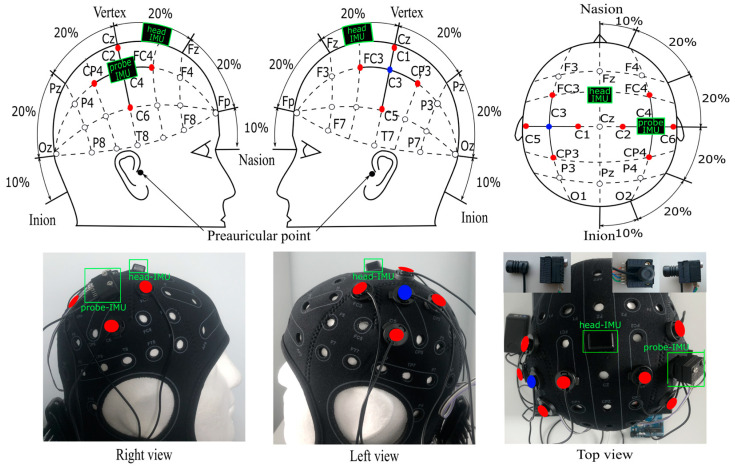
The positioning of optodes and IMUs. Positions follow the international transcranial positioning pattern 10/20 [37]. The fNIRS source and detector pairs are separated by 3 cm. Sources are shown in red and detectors in blue. The two IMU sensors are outlined in green. The head-IMU is at FCZ and the probe-IMU is on the top of detector C4.

**Figure 4 sensors-24-03173-f004:**
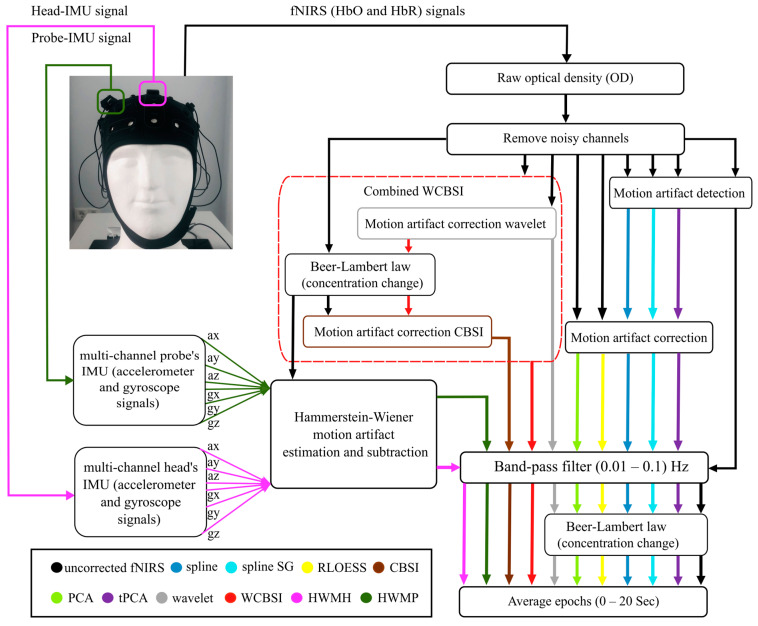
The processing streams for each MA correction technique are indicated by colored arrows. The processing of the uncorrected fNIRS signal is in black. The pipelines of the different correction approaches are shown in pink for HWMH, olive for HWMP, green for PCA, yellow for RLOESS, blue for spline, cyan for splineSG, and purple for tPCA, brown for CBSI, red for WCBSI, and grey for the wavelet.

**Figure 5 sensors-24-03173-f005:**
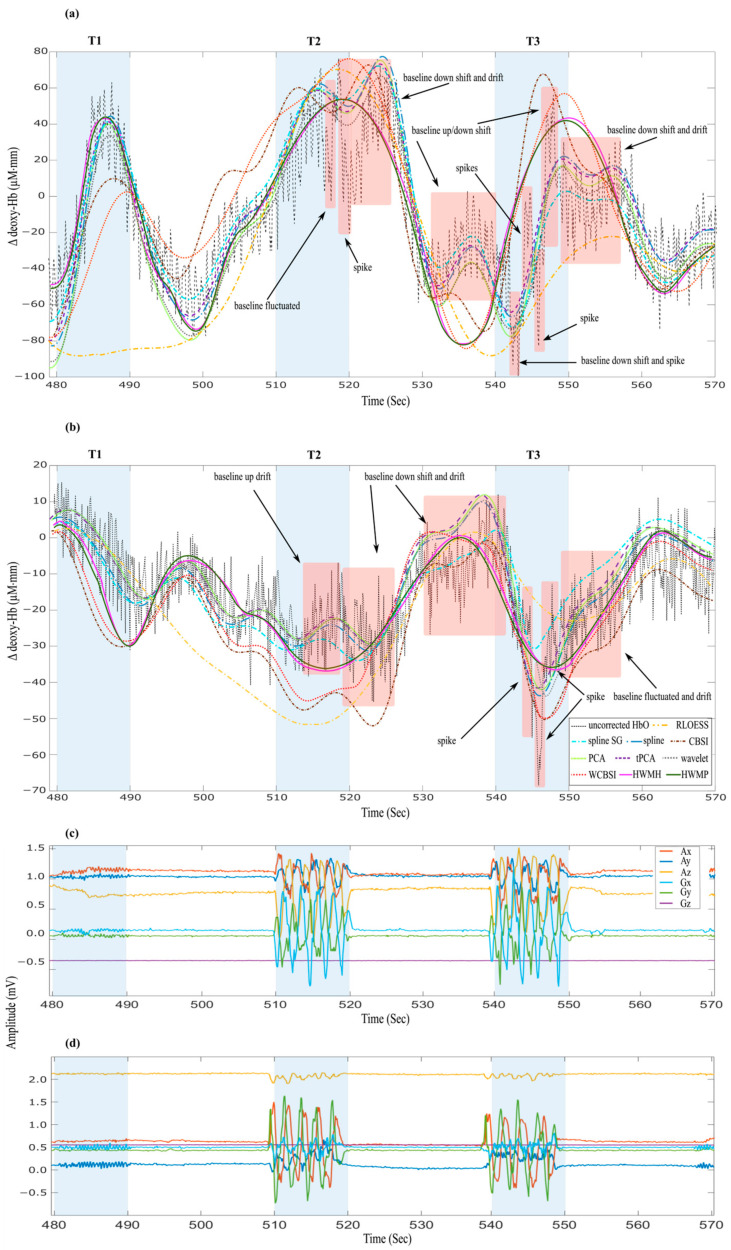
Effects of the different MA correction methods on various types of MAs are marked in red. The intervals of the three tasks (T1, T2, and T3) are marked in blue. Single-trial time series of (**a**) HbO, (**b**) HbR, (**c**) the probe-IMU, and (**d**) the head-IMU are shown. The uncorrected signal is shown as a black dashed line. Compared to all other methods, HWMH and HWMP correction (pink and green solid lines) exhibit remarkable efficacy in mitigating all types and magnitudes of MA for both hemoglobin signals during the T2 and T3 intervals. The corrected signals generated by the other MA corrections are shown as dashed lines with different colors: spline (blue), spline SG (cyan), RLOESS (orange), tPCA (purple), CBSI (brown), wavelet (grey), PCA (green), and WCBSI (red).

**Figure 6 sensors-24-03173-f006:**
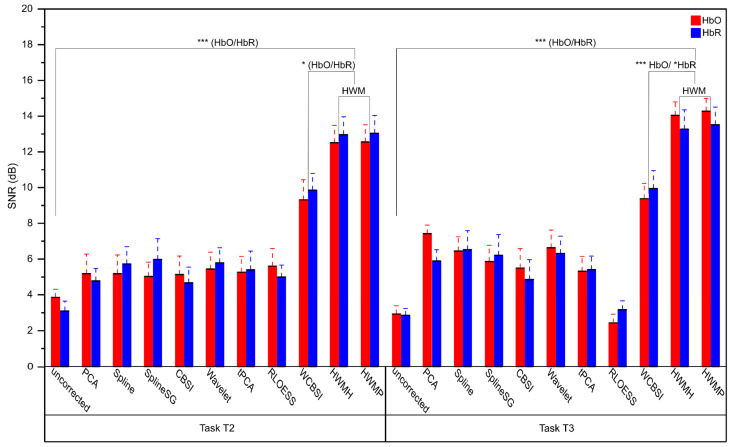
Mean signal-to-noise ratio (SNR) with standard errors after MA correction for the different correction methods plus the SNR of the uncorrected data. SNRs for HbO and HbR signals are shown in red and blue, respectively. The left panel shows SNRs for experimental tasks T2 (small head movements) and the right panel for T3 (large head movements), with a significance level of *p* < 0.001. Note that HWMH and HWMP have superior SNRs compared to all other methods tested. Three stars (***) indicate a significance level of *p* < 0.001, and one star (*) *p* < 0.05.

**Figure 7 sensors-24-03173-f007:**
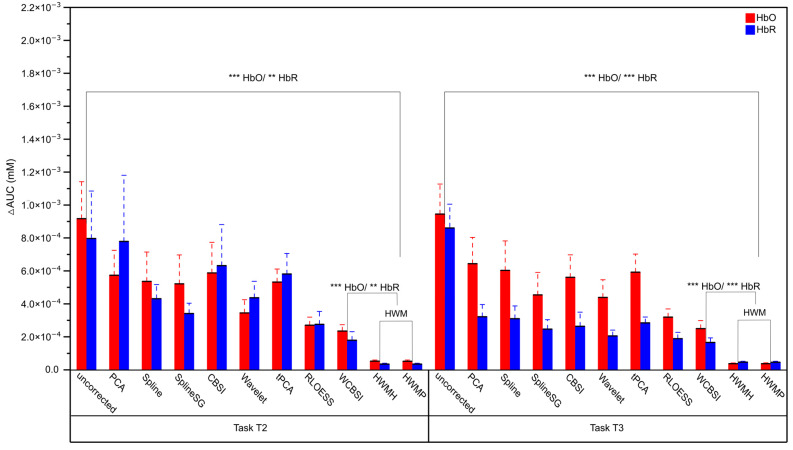
Mean area under the curve difference (ΔAUC) with standard errors after MA correction for the different correction methods plus the uncorrected data. ΔAUCs for HbO and HbR signals are shown in red and blue, respectively. The left panel shows results for experimental tasks T2 (small head movements) and the right panel for T3 (large head movements). Three stars (***) indicate a significance level of *p* < 0.001, and two stars (**) *p* < 0.01. Note that HWMH and HWMP corrections produce fNIRS signals with the smallest AUC deviation from the ground truth signal.

**Figure 8 sensors-24-03173-f008:**
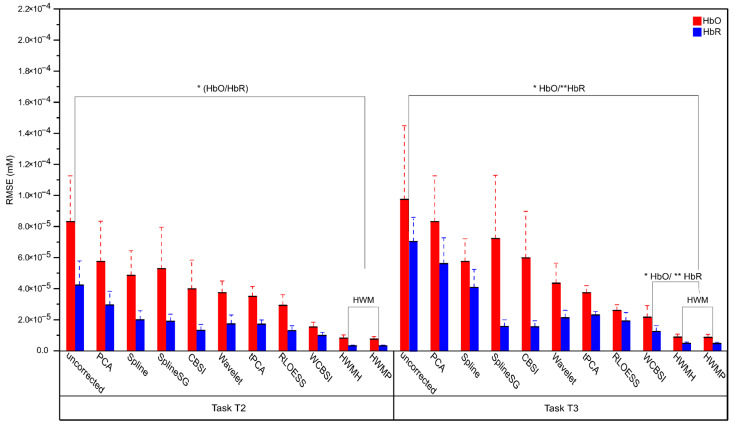
Mean RMSE with standard errors after MA correction for the different correction methods plus the uncorrected data. RMSEs for HbO and HbR signals are shown in red and blue, respectively. The left panel shows results for experimental tasks T2 (small head movements) and the right panel for T3 (large head movements). HWMH and HWMP corrections produce fNIRS signals, with the smallest RMSE from the ground truth signal. Two stars (**) indicate *p* < 0.01, and one star (*) *p* < 0.05.

**Figure 9 sensors-24-03173-f009:**
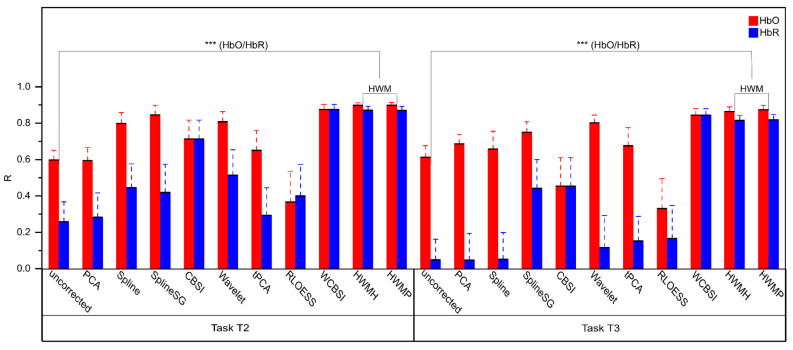
Mean Pearson correlation coefficient with standard errors after MA correction for the different correction methods plus the uncorrected data. Rs for HbO and HbR signals are shown in red and blue, respectively. The left panel shows results for experimental tasks T2 (small head movements) and the right panel for T3 (large head movements). Three stars (***) indicate a significance level of *p* < 0.001. Note that HWMH, HWMP, and WCBSI produce MA-corrected fNIRS signals with the highest correlation with the ground truth signal.

**Figure 10 sensors-24-03173-f010:**
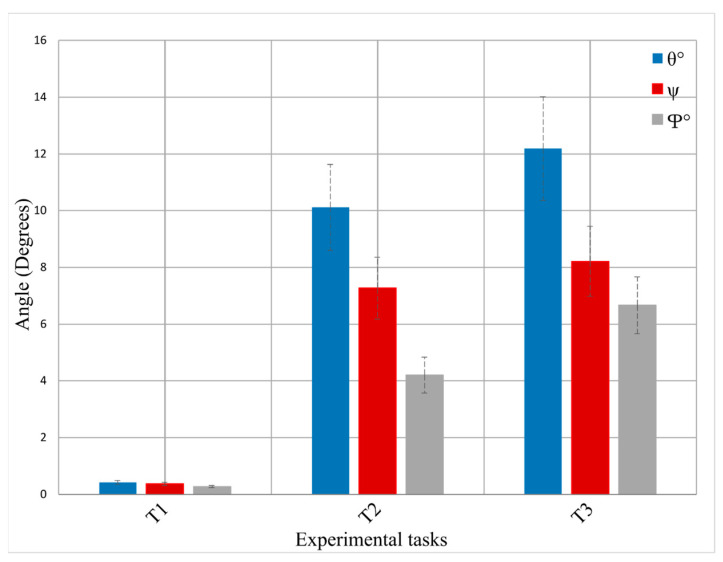
Mean head rotation angles in the three experimental conditions: pitch (θ, blue) is the rotation angle around the *x*-axis, roll (ψ, red) is the rotation angle around the *y*-axis, and yaw (φ, grey) is the rotation angle around the *z*-axis. Error bars indicate the standard error of the mean.

**Figure 11 sensors-24-03173-f011:**
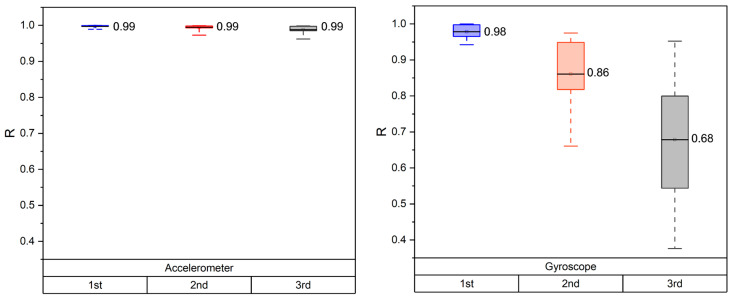
The correlation coefficients of the first three CCA variables are denoted by the colors blue, red, and grey, respectively, for the accelerometer and gyroscope measurements. These coefficients represent the relationships between the head-IMU and probe-IMU in three degrees of freedom.

## Data Availability

The dataset was made open access for future development and evaluations at https://www.doi.org/10.17605/OSF.IO/U3F89 (accessed on 12 May 2024).

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
