# Peer review of "Hammerstein–Wiener Motion Artifact Correction for Functional Near-Infrared Spectroscopy: A Novel Inertial Measurement Unit-Based Technique"

_sensors, 2024, doi:10.3390/s24103173_

Round 1
Reviewer 1 Report
Comments and Suggestions for Authors
This study proposes an IMU-based technique to deal with motion artifact correction for fNIRS. The topic is interesting. The paper is well organized with clear motivation and good novelty. However, the following issues should be addressed before I recommend it for publication:
(1) Seventeen healthy participants were enrolled in the experiment. How to control the representative of participants in age, gender and etc?
(2) The method cannot realize real-time data processing and correction and thus hinders its application.
(3) The paper needs minor revision.
Reviewer 2 Report
Comments and Suggestions for Authors
The work presents a technique to compensate for head movements in fNIRS recording. The technique uses inertial sensors combined with a Hammerstein-Wiener estimator.
The work is interesting and novel, and the results are valuable. Although the manuscript is generally clear, some methodological issues need to be clarified before this work is accepted.
It is important to highlight that the authors provide the experimental data and the code used in the Arduino to record the IMU signals.
The following is the list of major issues that, in my opinion, need to be addressed:
* Major Issues
- Eqs. (2), (3), and (4) describe the HWM. However, no details are given about how you designed the model. In particular, there needs to be an explanation about how you get the value of all the model parameters and which values you used. Considering that the use of the HWM is one of the key contributions of this work, it is paramount that all the details about how to use this model are very clear. For these reasons, I suggest expanding the description of the model and giving all the necessary details for a reader to reproduce your results.
- Clarify the explanation of Eq. (2): The equation uses Q1, but the text mention Q; the equation uses i, the text uses I, what do you mean by "kernel's diagonal"? Spell out the dimension of x, etc.
- In line 135, what do you mean "If the MAs are larger than the brain signal"? Larger in magnitude? Please clarify.
- Give more details about the acquisition of the IMU signals: What sampling frequency did you use? How did you sincronize the fNIRS data with the IMU signals.
- In line 249, spell out the order and type of the band-pass filter.
- In Eq. (8), explain how AUC_HRF and AUC_{\hat{HRF}} are calculated.
The following are some minor issues.
* Minor Issues
- Be consistent with the parenthesis. Figure 1 uses square parenthesis (y[t], etc.), while the text uses rounded parenthesis (y(t), etc.).
- Line 272: Spell out the meaning of the HRF abbreviation.
Reviewer 3 Report
Comments and Suggestions for Authors
As a whole, the proposed manuscript seems to be comprehensive research, as done experiments are described accurately and in details. Figures are informative enough. Text is clear, only minor text corrections are possible.
Some drawbacks of the proposed paper are mainly refer to the description of the experimental scheme.
1. Unfortunately, the detailed description of the used method does not contain the brief description of the basic idea of the experiment, involving measurements of the final amount of oxyhemoglobin (HbO) and the deoxyhemoglobin (HbR) in the irradiated human head skin. As such a complex object contains various interacting biological environments (epidermis, hair, blood vessels, skull bones etc.), then it is necessary to explain briefly, what set of real physical/chemical processes may be involved by the scattered IR signal. For some tasks this set can limit the application of the proposed data acquisition and procession method.
2. In sec.2 firstly comment the basic idea of Hammerstein- Wiener model (HWM), before the commentary to nonlinear blocks m1, m2, and the linear one h(𝜏)
3. The idea and the application of Beer-Lambert law are also to be explained more clearly in the text.
4. Comment, please, if the described set of experiments correlate somehow to well-known methods of IR laser stimulation and acupuncture?
5. What factors influence on the choice of the distance between the irradiator and the sensor, and if IR signal scattering factors are substantial here?
6. What was the role of the frequency of hand tapping? If it`s choice can somehow optimize output signals?
As a whole, the proposed manuscript can be recommended for publishing in Sensors with minor text corrections due to remarks given above.
Comments on the Quality of English LanguageMinor corrections of several phrases.
Reviewer 4 Report
Comments and Suggestions for Authors
The authors proposed a novel IMU-Based technique for motion artifact correction in fNIRS. They went on onto describing current well-known MA removal techniques such as tPCA, PCA, and CBSI. In addition, authors included the presentation of smoothing methods and linear combinations of direct movement measurements from accelerometers and gyroscopes.
As pointed out, authors evaluated the performance of their suggested non-linear MA correction method on empirical data and went on to performing the comparison among eight different popular methods. Also, the empirical execution of the experimental conditions presented by the authors, delivered a dataset which aspects were assessed quantitatively and qualitatively.
For the comparison and evaluation among the methods, authors used four basic metrics: Area Under the Curve (AUC), Root Mean Square Error (RMSE), Pearson correlation coefficient (R), and Signal to Noise Ratio (SNR). In addition, authors performed a paired t-test on all metrics to follow up on the significant differences in the performance of each tested algorithm.
Regarding the participants, authors included seventeen healthy participants with an average age of 29 ± 4 years. Authors also included a balanced number of males and females, with a detailed description of the hair colours and pre-existent neurological and psychological disorders.
Thanks to the authors for such and interesting research, and I would like you addressing the following:
· The number of participants: how did the authors come up with this number? (17) is it enough to argue validity of your tests?
· Line [345]: why the authors selected these metrics and no others?
· Line [345]: Please check grammar (colons seem to be missing after “quantify”)
· Sentence starting in line [349]: I do not understand the argument as to why authors decided to perform two statistical comparisons only, as “to reduce the number of significant tests” does not seem appropriate.
· I have concerns on the selection and application of the significance tests among the metrics used. Although paired t-tests seem appropriate for a single comparison between HbO vs HbR signals on the same subject, the purpose of the study (it seems) is to grasp differences among the ground truth, T1, and T2 (and perhaps features), situations where using an uncorrected pairwise comparison is not appropriate.
· Authors do really need to have a look to the interpretation of the Pearson’s correlation coefficient. What is stated in the sentence starting in line 407 is incorrect, and therefore all conclusions that followed its misinterpretation (especially what is stated in lines 415 to 421.)
Comments on the Quality of English LanguageMinor grammatical errors.
Reviewer 5 Report
Comments and Suggestions for Authors
This study addresses motion artifacts in functional Near-Infrared Spectroscopy (fNIRS) experiments, proposing a novel application of the non-linear Hammerstein-Wiener model to mitigate these artifacts. They evaluate this approach using IMU sensors on both the participant's head and the fNIRS probe. Their approach achieves significant improvement in signal-to-noise ratio (SNR), comparable to using IMU sensors.
Any claims mentioning "significant improvement" requires rigorous statistical testing. These tests are either missing entirely or improperly applied. The authors should work with a statistician to validate the claims and describe the statistical methods used in the work properly.
Comments on the Quality of English Language
The discussion section can be shortened.
Round 2
Reviewer 2 Report
Comments and Suggestions for Authors
The authors have addressed all the issues. I recommend accepting the paper.
Reviewer 4 Report
Comments and Suggestions for Authors
After conducting a thorough evaluation of this new version of the paper, I am pleased to acknowledge the authors' efforts in implementing the suggested corrections and updates to their research, in line with the feedback provided by the reviewers. Their responsiveness to constructive criticism is commendable and greatly appreciated.
Comments on the Quality of English LanguageNo major English issues detected.
Reviewer 5 Report
Comments and Suggestions for Authors
The authors have addressed all the comments and the paper can now be accepted.